# Cognitive Profiling of Children and Adolescents with ADHD Using the WISC-IV

**DOI:** 10.3390/bs15091279

**Published:** 2025-09-18

**Authors:** Megan Rosales-Gómez, Ignasi Navarro-Soria, Manuel Torrecillas, María Eugenia López, Beatriz Delgado

**Affiliations:** 1Department of Developmental Psychology and Didactics, Faculty of Education, University of Alicante, 03690 Alicante, Spain; megan.rosales@ua.es (M.R.-G.); manuel.torrecillas@ua.es (M.T.); beatriz.delgado@ua.es (B.D.); 2Department of Experimental Psychology, Cognitive Processes and Speech Therapy, Universidad Complutense de Madrid, 28040 Madrid, Spain; mariaelo@ucm.es; 3Centre for Cognitive and Computational Neuroscience, Universidad Complutense de Madrid, 28040 Madrid, Spain

**Keywords:** ADHD, cognitive profile, executive functions, neuropsychological assessment, WISC-IV

## Abstract

Attention Deficit Hyperactivity Disorder (ADHD) is a prevalent neurodevelopmental disorder characterised by cognitive and behavioural impairments. This study aimed to identify cognitive patterns associated with ADHD in a sample of 719 children and adolescents (363 with ADHD and 356 controls) assessed using the Wechsler Intelligence Scale for Children—Fourth Edition (WISC-IV). Compared to controls, the clinical group exhibited significantly lower scores in the Working Memory Index (WMI), Processing Speed Index (PSI), and Cognitive Proficiency Index (CPI). No significant group differences were found in Verbal Comprehension (VCI) or Perceptual Reasoning (PRI) after controlling for age and sex. Factorial MANOVA results revealed that WMI, PSI, and CPI deficits remained stable across age groups and were more pronounced in males. Females with ADHD outperformed males in PSI. A binary logistic regression model including WISC-IV core indices classified VCI, PRI, WMI, and PSI with a Nagelkerke R^2^ of 0.44 as significant predictors of group membership, indicating that lower scores in WMI and PSI, and higher scores in VCI and PRI, increased the likelihood of ADHD classification. These findings reinforce the use of the WISC-IV as a complementary tool in the cognitive characterisation and clinical assessment of ADHD in youth.

## 1. Introduction

Attention-Deficit/Hyperactivity Disorder (ADHD) is among the most prevalent neurodevelopmental disorders, affecting approximately 6% to 8% of children worldwide ([6]). According to the American Psychiatric Association ([5]), ADHD is characterised by three core dimensions: inattention, hyperactivity, and impulsivity. These symptoms are grouped into three clinical presentations: predominantly inattentive, predominantly hyperactive/impulsive, and the combined presentation. However, regardless of the subtype, ADHD has a significant impact on both academic performance and children’s socio-emotional adjustment ([47]). Therefore, early identification and appropriate intervention are critical for addressing the difficulties associated with the disorder, whereas delayed or inaccurate diagnosis may lead to long-term negative consequences in a child’s development.

Within the developmental psychopathology framework, the deficits often found in ADHD reflect atypical developmental pathways ([17]). These difficulties in attention and executive control can be understood as disruptions in the self-regulatory systems of the mind, which normally develop through the interaction between brain development and environmental demands, and when disrupted, difficulties like those observed in ADHD may appear ([12]). A recent meta-analysis by [23] ([23]) highlights how the distinct symptoms of ADHD are associated with differentiated outcomes in the functional development of children and adolescents. Inattention is strongly linked to academic difficulties, low self-esteem, poor occupational outcomes, and reduced overall adaptability; in contrast, hyperactivity/impulsivity symptoms are associated with peer rejection, aggression, and an increased risk of accidents ([23]). In schools, ADHD is often identified because of learning difficulties that begin to present themselves when children reach school age, including challenges in remaining seated for extended periods, sustaining attention, and completing tasks that require higher-order cognitive functions such as planning, mathematical calculation, and reading ([10]; [26]). These challenges frequently lead to school maladjustment ([40]). In addition to academic performance, emotional regulation and peer relationships are also negatively affected. When ADHD persists into adulthood, it may result in lower socioeconomic status, occupational instability, and other difficulties in daily life, particularly in the absence of treatment ([3]).

The diagnosis of ADHD is determined according to the criteria established in the Diagnostic and Statistical Manual of Mental Disorders, Fifth Edition (DSM-5), which requires the presence of at least six symptoms in two or more settings, such as at home, in social situations, or at school with an onset before age 12 ([5]). Additionally, the symptoms must not be better explained by another mental disorder. According to the DSM-5, individuals with ADHD typically experience marked difficulties with attention, organisation, and/ or hyperactivity. Attentional and organisational problems include an inability to maintain focus on tasks, appearing not to listen, and frequently losing items necessary for daily activities, with a level of severity that is inconsistent with the individual’s age or developmental level. On the other hand, hyperactivity/impulsivity is characterised by excessive activity, persistent restlessness, difficulty remaining seated, frequent interruptions of others, and problems waiting one’s turn ([5]).

Neurobiological evidence supports these diagnostic criteria, revealing structural and functional alterations in regions involved in executive functions and the regulation of the dopaminergic system, including the dorsolateral prefrontal cortex, anterior cingulate cortex, basal ganglia, and limbic system ([1]; [8]; [23]; [36]). These findings are directly related to the deficits observed in executive functions, one of the most affected cognitive domains in individuals with ADHD ([38]; [44]; [47]; [51]). Executive functions (EF) refer to a set of higher-order mental processes that are essential for sustained attention, deliberate decision-making, and goal-directed behaviour, particularly in situations where automatic or instinctive responses are insufficient or inappropriate ([18]).

In particular, ADHD is associated with difficulties in key EF domains such as inhibitory control, cognitive flexibility, and working memory (WM), as well as related domains including visuospatial skills, sustained attention (vigilance), and processing speed (PS) ([32]; [39]; [25]; [47]; [51]). WM is essential for manipulating information over short periods, enabling the retention and use of information in real time ([18]). Previous studies have shown that this capacity is commonly impaired in children with ADHD, negatively affecting academic performance and the ability to complete complex tasks ([16]; [38]; [44]). PS, which refers to the speed and accuracy with which cognitive tasks are carried out, is another domain that is notably found to be affected in ADHD ([19]). Slowness in PS interferes with the ability to respond effectively to environmental demands and influences both learning and social interactions ([31]; [56]).

These neurobiological alterations are directly correlated with the cognitive deficits described in inhibitory control, WM, PS, planning, and organisation ([23]). These deficits, in turn, manifest clinically as symptoms of inattention and hyperactivity/impulsivity in the daily lives of individuals with ADHD. Both the observable symptoms and measurable cognitive impairments appear to vary in severity across development, with some studies reporting improvements in cognitive functioning with age ([42]). A meta-analysis described larger effect sizes in children and adolescents compared to adults when assessing deficits across multiple cognitive domains ([23]; [41]). This is also illustrated by another meta-analysis involving more than 8200 children and adolescents with ADHD, which found deficits in WM decreased with age ([43]; [23]). These findings are consistent with earlier studies indicating that hyperactivity/impulsivity symptoms tend to diminish over time, whereas inattention symptoms remain relatively stable ([6]; [55]).

Moreover, some studies suggest that the clinical and cognitive manifestations of ADHD may vary according to sex. Females tend to exhibit greater difficulties in sustained attention, whereas males more frequently present hyperactive/impulsive symptoms ([1]). As a result, it has been suggested that females with ADHD are more likely to be underdiagnosed compared to males with the disorder ([9], as cited in [6]). However, a series of more recent meta-analyses found no significant differences between sexes in any ADHD symptom domain ([33]). Nevertheless, the prevalence of ADHD remains approximately twice as high in males than in females ([6]).

In clinical practice, diagnosis is primarily based on behavioural symptoms, relying largely on clinical observation using DSM-5 diagnostic criteria, clinical interviews, and behavioural rating scales and questionnaires ([1]). According to the DSM-5, in children, inattention is manifested through difficulties in maintaining concentration, careless mistakes, failure to follow instructions, disorganisation, avoidance of tasks requiring sustained mental effort, frequent forgetfulness in daily activities, and being easily distracted by external stimuli ([5]). Hyperactivity-impulsivity, on the other hand, is characterised by excessive restlessness, difficulty remaining seated, inappropriate behaviours such as running or climbing at inappropriate times, difficulty engaging in quiet play, excessive talking, impulsive responding, impatience, and a tendency to interrupt others ([5]). However, the expression of these symptoms can also be shaped by culture. Values, beliefs, and practices may either suppress or amplify certain behaviours, while the degree of tolerance from others, such as parents, teachers, and peers, influences which behaviours are considered problematic or inappropriate ([11]).

The most commonly used diagnostic questionnaires include tools such as the ADHD-Rating Scale DSM-5 for parents and teachers (ADHD-RS) ([21]) and symptom evaluation checklists such as the Child Behaviour Checklist (CBCL) ([2]). These standardised instruments are designed to assess symptoms of inattention, hyperactivity, and impulsivity, and are completed by parents and teachers. However, the results obtained through both clinical observation and these questionnaires may be influenced by the subjective biases of observers ([15]; [22]; [34]; [37]), whose responses can vary depending on their personal interpretation of what constitutes hyperactivity or inattention, as well as on contextual factors such as expectations, the child’s environment, and the child’s sex. Furthermore, each cultural group will differ in appropriate and inappropriate behaviour, having different thresholds that vary in accordance with what is regarded as desirable or unacceptable for each individual in their own and different context ([11]). These diagnostic questionnaires may, therefore, fail to fully capture the underlying cognitive deficits characteristic of ADHD. According to [7] ([7]), one of the main limitations of the current diagnostic system is its reliance on subjective reports from parents and teachers, who are not always reliable informants.

In addition, the diagnosis of ADHD is further complicated by its high comorbidity with other disorders, including behavioural disorders (52%), anxiety (33%), depression (17%), and autism (14%) ([1]). Symptoms such as disinhibition, impulsivity, emotional dysregulation, and executive function deficits often overlap with manifestations of other neurodevelopmental conditions, anxiety disorders, oppositional defiant disorder (ODD) and mood disorders ([5]; [23]; [30]). This symptom overlap complicates identification, as the signs are not exclusive to ADHD and may be perceived differently depending on the context or the observer. Due to the complexity of establishing an accurate diagnosis, there is a risk of overdiagnosis, which may lead to unnecessary medication prescriptions and potential adverse effects in individuals who do not meet the criteria for the disorder ([1]). At the same time, underdiagnosis or misdiagnosis also pose a risk, potentially resulting in inadequate or absent treatment. Individuals with untreated ADHD are at increased risk of developing adverse outcomes, including substance abuse, suffering from traffic accidents or unintentional injuries, depression, anxiety, and even suicide ([4]; [23]; [29]).

These findings stress the importance of employing diagnostic tools that minimise biases derived from subjective interpretation by incorporating complementary tools, such as cognitive assessments, to more accurately identify the neuropsychological deficits underlying observable behaviour. The integration of parent and teacher reports provides complementary perspectives on the child’s behaviour across different settings, thereby improving diagnostic accuracy ([48]). However, [28] ([28]) reported that classification model accuracy was higher when clinical rating scales were combined with neuropsychological tests (70.44%) than when either method was used independently. This highlights the need to supplement behavioural rating scales with cognitive assessments, as the latter allow for the identification of ADHD-specific deficits. Combining these tools thus yields a more comprehensive and accurate understanding of the child’s cognitive profile and associated difficulties.

The Wechsler Intelligence Scale for Children—4th Edition (WISC-IV; [53]) is one of the most widely used tools for neuropsychological assessment in children. It has been validated in Spain and has proven effective for evaluating cognitive functioning in children aged 6 to 16 years ([24]). The test is designed to assess various cognitive abilities, including verbal comprehension (VC), perceptual reasoning (PR), WM, and PS, which are referred to as the core index scores. Previous studies on ADHD with the WISC-IV, such as [25] ([25]), [38] ([38]), and [50] ([50]), also compared overall IQ and specific WISC-IV indices between children with ADHD and control groups; they consistently found statistically significant deficits in WM and PS in children with ADHD, while VC and PR tended to be preserved. However, the samples in these studies were smaller than the present one, ranging from 126 to 189 participants.

In line with developmental psychopathology, research suggests that individuals with ADHD are not characterised by a general reduction in overall intellectual ability, but rather by deficits in specific domains such as WM and EF ([17]). Thus, the main objective of the present study is to identify cognitive patterns in children and adolescents diagnosed with ADHD through the assessment of the core index scores of the WISC-IV, examining its utility as a complementary tool in the diagnostic process beyond traditional behavioural criteria in a large sample of 719 children and adolescents. Specifically, the study aims to: (1) evaluate differences in the cognitive profiles obtained through the WISC-IV between children and adolescents with an ADHD diagnosis and a control group without ADHD; (2) explore the influence of age and sex on cognitive outcomes in both groups; and (3) examine the potential utility of WISC-IV index scores as predictors in the identification of children and adolescents with ADHD. Accordingly, the study was guided by four research questions: RQ1: Do children and adolescents with ADHD differ from controls across the WISC-IV indices? RQ2: How do age and sex influence cognitive outcomes in both groups? RQ3: Can WISC-IV scores predict group membership (ADHD vs. control)? RQ4: Are there significant interaction effects among diagnosis, sex, and age in explaining cognitive performance? Based on prior literature, it was hypothesised that (H1) children and adolescents with ADHD would show significantly lower scores in WMI and PSI, but no group differences in VCI or PRI; (H2) that females with ADHD would outperform males in PSI; and (H3) that cognitive deficits in WMI and PSI would diminish with age, with adolescents showing fewer deficits as their development progresses.

## 2. Materials and Methods

### 2.1. Design

The present study employed a comparative, cross-sectional design aimed at identifying cognitive patterns in children and adolescents diagnosed with ADHD (without comorbidities) and comparing them to a control group of neurotypical peers. The methodological framework was based on data provided by the Research Group in Developmental Psychology and Criminology (GIPEC) at the University of Alicante, with the objective of verifying and expanding upon previously reported findings using a broader and more diverse sample. A quantitative approach was adopted, involving the administration of the WISC-IV and statistical analyses designed to identify significant differences between groups and contribute to the cognitive characterisation of children and adolescents with ADHD.

### 2.2. Participants

The sample consisted of 719 children and adolescents of both sexes, aged between 6 and 16 years. Participants were divided into two groups: a clinical group diagnosed with ADHD (*n* = 363) and a control group without a diagnosis of ADHD or other disorders (*n* = 356). Participants in the clinical group were recruited from Child and Adolescent Mental Health Units, psycho-pedagogical counselling centres, and family associations of children with ADHD located in the provinces of Murcia, Alicante, and Valencia (Spain). The control group included students from schools in the same provinces, ensuring adequate representativeness and comparability between samples.

The inclusion criteria replicated those used in [38] ([38]) and included: (a) a clinical diagnosis of ADHD confirmed by a qualified mental health professional using DSM-5 criteria; (b) an intelligence quotient (IQ) of 80 or above; (c) absence of severe mental disorders or neurological diseases, as determined by a clinical professional; and (d) a minimum 48 h suspension of pharmacological treatment prior to assessment. For the control group, the same criteria were applied, excluding the ADHD diagnosis. Additionally, all participants were required to meet non-clinical cut-off scores on the Strengths and Difficulties Questionnaire (SDQ; [27]) to ensure the absence of significant emotional or behavioural difficulties.

Exclusion criteria included the presence of learning disorders, autism spectrum disorder, intellectual disability, acquired brain injury, epilepsy or other neurological conditions, ongoing psychostimulant medication use without appropriate suspension, and incomplete or inconsistent data in the neuropsychological assessments. After applying the inclusion and exclusion criteria, 137 were excluded due to not meeting diagnostic requirements, presenting comorbid conditions, failing to reach the minimum IQ threshold, and providing incomplete or inconsistent data. Thus, the final sample of 719 participants was obtained.

### 2.3. Assessment Tools

To establish the diagnosis of ADHD, several standardised instruments were employed. The Strengths and Difficulties Questionnaire (SDQ; [27]), a 25-item screening tool completed by parents and teachers, was used to assess emotional and behavioural difficulties in both groups. The ADHD-Rating Scale DSM-IV for Parents and Teachers (ADHD-RS-IV; [20]), validated in the Spanish population ([45]; [52]), was used to measure the severity of inattention and hyperactivity/impulsivity symptoms. In addition, the Conners’ Rating Scale, Third Edition (short version) for parents and teachers ([14]) was administered to assess behavioural and symptomatic profiles commonly associated with ADHD ([35]). Symptom information was further explored through the Parental Account of Children’s Symptoms (PACS; [49]), a semi-structured interview conducted with parents to corroborate and clarify reported behaviours.

In cases where the information provided by parents was insufficient, ambiguous, or inconsistent, an additional interview was carried out with the child’s or adolescent’s teacher to obtain complementary data.

Furthermore, the Spanish version of the Wechsler Intelligence Scale for Children—Fourth Edition (WISC-IVspa; [54]) served as the instrument for the neuropsychological assessment of participants’ cognitive abilities. This scale is widely recognised for its strong psychometric properties and has been extensively applied in ADHD research to provide a valid and reliable measurement of multiple cognitive domains. The WISC-IV was standardised on a national sample of 1590 Spanish children and adolescents and demonstrated reliability coefficients ranging from 0.86 to 0.95 ([54]; [24]). It evaluates a range of cognitive functions through four main indices: Verbal Comprehension Index (VCI), Perceptual Reasoning Index (PRI), Working Memory Index (WMI), and Processing Speed Index (PSI). These indices assess, respectively, verbal reasoning, visuospatial perceptual reasoning, the ability to actively manipulate short-term information, and cognitive speed and accuracy in visual and motor tasks. From these, two composite scores are derived: the General Ability Index (GAI), which combines VCI and PRI to estimate general intellectual ability, and the Cognitive Proficiency Index (CPI), which integrates WMI and PSI to assess efficient cognitive performance. In this study, the ten core subtests that make up the four primary indices were administered.

### 2.4. Procedure

The data collection procedure consisted of three key phases. First, an initial diagnostic assessment was conducted by clinical professionals in healthcare centres. During this phase, standardised diagnostic instruments were administered, including the SDQ, ADHD-RS-IV, Conners, and PACS. In cases where the information provided by parents was ambiguous, incomplete, or inconsistent, an additional interview with the teachers was conducted.

In the second phase, a rigorous data cleaning process was carried out to ensure validity. This involved verifying normative score ranges, checking internal consistency among instruments, detecting missing or contradictory values, and reviewing all scale scores. Cases with incomplete or inconsistent data were excluded to maintain the reliability of subsequent analyses.

Finally, participants underwent cognitive evaluation using the Spanish version of the WISC-IV. This assessment was conducted individually by trained psychologists in a controlled and appropriate environment. Informed consent was obtained from parents or legal guardians before participation. The study received approval from the Ethics Committee of the University of Alicante (UA-2023-06-30_1) and was conducted in accordance with the principles of the Declaration of Helsinki.

### 2.5. Statistical Analysis

All statistical analyses were conducted using IBM SPSS Statistics, version 27. The procedures included descriptive analyses, independent samples t-tests, binary logistic regression, and a factorial MANOVA, each performed to address one or more of the research questions described earlier. To begin, the normality of the distributions was assessed using the Kolmogorov–Smirnov test with a significance level set at α = 0.05. This test was applied to all continuous variables of interest, such as the WISC-IV index scores. The results indicated that all variables met the assumption of normality, as shown by *p*-values greater than 0.05. This justified the use of parametric statistical methods in the subsequent analyses.

Descriptive statistics were computed (means and standard deviations for continuous variables; frequencies and percentages for categorical variables), and a chi-square test was also conducted to examine whether the distribution between the clinical group and the control group was homogeneous in terms of sex and age.

Furthermore, in order to answer RQ1, independent samples t-tests were applied to compare the ADHD and control groups across each WISC-IV index, including both primary (VCI, PRI, WMI, PSI) and composite indices (FSIQ, GAI, CPI). To further address RQ2 the t-test was also applied within each group to explore potential differences based on sex and age. Age was dichotomised into two categories (children aged 6 to 11 vs. adolescents aged 12 to 16) to reflect key developmental stages in childhood and adolescence. Cohen’s d was calculated to estimate effect sizes, following conventional interpretation guidelines ([13]).

Next, to examine RQ3, a binary logistic regression was conducted to assess the predictive utility of the WISC-IV indices (VCI, PRI, WMI, PSI) in classifying participants into clinical or control groups. Goodness of fit was evaluated using the Nagelkerke R^2^ coefficient.

Lastly, a multivariate analysis of variance (MANOVA) was conducted to examine RQ4, which focused on the main and interaction effects of group (ADHD vs. control), sex (males vs. females), and age group (children vs. adolescents) on the WISC-IV cognitive indices.

### 2.6. Data Availability

The data supporting the findings of this study are not publicly available due to ethical restrictions concerning the confidentiality and protection of the participating minors. Access to the data may be considered upon request to the corresponding author, subject to relevant ethical and institutional approval.

## 3. Results

### 3.1. Descriptive Characteristics of the Sample

The total sample consisted of 719 participants, with a distribution of 547 males (76.1%) and 172 females (23.9%). The mean age of the participants was 8.51 years (SD = 2.50), ranging from 6 to 16 years. Of the total sample, 85.1% were children (aged 6–11) and 14.9% were adolescents (aged 12–16). In terms of group distribution, 49.5% of the participants belonged to the control group (*n* = 356) and 50.5% to the clinical group diagnosed with ADHD (*n* = 363). See Table 1.

The chi-square test showed a statistically significant association between sex and group membership (χ^2^(1) = 12.030, *p* = 0.001), indicating a higher proportion of males in the clinical group. No significant differences were found between sex and age group (χ^2^(1) = 1.170, *p* = 0.279). However, a significant association was observed between age group and clinical and control group (χ^2^(1) = 15.891, *p* < 0.001), with a lower proportion of adolescents in the ADHD group.

### 3.2. Comparison Between ADHD and Control Groups

Independent samples t-tests were conducted to compare WISC-IV index scores between the control and clinical (ADHD) groups. The results revealed statistically significant differences across all evaluated indices (Table 2).

The clinical group showed significantly lower scores in WMI, PSI, FSIQ, and CPI, with medium to large effect sizes for WMI (d = 0.85), PSI (d = 0.63), and CPI (d = 0.91), and a small effect size for FSIQ (d = 0.23). In contrast, the ADHD group obtained slightly higher means in VCI and PRI, although the effect sizes were small (d = 0.21 and 0.22, respectively).

As shown in Figure 1, the control group displays a relatively consistent cognitive profile across all WISC-IV indices. In contrast, the ADHD group shows a noticeable decline in specific cognitive domains. While scores in VCI and PRI are slightly elevated, there is a marked decrease in performance beginning with the WMI and continuing through the PSI and the CPI. This descending trend shows the pronounced difficulties in cognitive efficiency experienced by the ADHD group, particularly in tasks requiring WM and PS.

### 3.3. Sex-Based Comparisons

Within the clinical group, statistically significant sex differences were observed in PSI (*p* < 0.001, d = 0.525) and CPI (*p* = 0.010, d = 0.349), with females with ADHD obtaining higher scores. No significant differences were found for VCI, PRI, WMI, or GAI. No significant sex differences were found in the control group (see Table 3).

### 3.4. Age-Based Comparisons

Children (6–11 years) with ADHD scored significantly higher than adolescents (12–16 years) on VCI, PRI, FSIQ, and GAI, with small effect sizes. No significant differences were observed in WMI, PSI, or CPI. In the control group, children outperformed adolescents on VCI, FSIQ, and GAI, although effect sizes were small. Detailed results are shown in Table 4.

### 3.5. Logistic Regression

A binary logistic regression analysis was conducted to determine the predictive capacity of the WISC-IV indices in classifying participants into the clinical or control group. The overall model was significant (χ^2^(4) = 290.77, *p* < 0.001), and reliably distinguished between groups. Including the four core cognitive domains (VCI, PRI, WMI, and PSI), the model explained 44.4% of the variance (Nagelkerke R^2^ = 0.444) and correctly classified 78% of cases, with all predictors reaching statistical significance (*p* < 0.001). The results are shown in Table 5.

These results indicate that lower scores on WMI and PSI and higher scores on VCI and PRI significantly increase the likelihood of belonging to the clinical ADHD group. To address the unequal distribution of sex and the slight age differences between groups, an additional logistic regression was conducted controlling for these variables. The results did not change, confirming that the predictive value of the WISC-IV indices is robust and not explained by demographic imbalances.

### 3.6. Multivariate Analysis of Variance (MANOVA)

A factorial MANOVA was conducted to examine the main effects of sex (male vs. female), diagnostic group (ADHD vs. control), and age group (children vs. adolescents) on the seven WISC-IV indices. The multivariate results revealed several statistically significant main effects. Table 6 summarises the univariate F-values, *p*-values, and partial eta-squared (η^2^) values for each factor on each index.

#### 3.6.1. Effect of Sex

Significant differences by sex were observed in WMI (F(1, 711) = 4.382, *p* = 0.037, η^2^ = 0.006), PSI (F(1, 711) = 14.902, *p* < 0.001, η^2^ = 0.021), and CPI (F(1, 711) = 12.962, *p* < 0.001, η^2^ = 0.018), indicating that females outperformed males in working memory and processing speed indices, although all with small effect size. No significant sex differences were found for VCI, PRI, FSIQ, or GAI. In addition, a statistically significant interaction was observed between sex and diagnostic group when controlling for age, specifically in PSI scores (F(1, 711) = 4.60, *p* = 0.032, η^2^ = 0.006). As shown in Figure 2, the significant slope difference shows females with ADHD scored higher in PS than males with ADHD, also aligning with the previous Student’s t-test.

#### 3.6.2. Effect of Diagnostic Group

The diagnostic group had significant effects on several indices. Children with ADHD showed lower scores on WMI (F(1, 711) = 41.476, *p* < 0.001, η^2^ = 0.055), PSI (F(1, 711) = 18.619, *p* < 0.001, η^2^ = 0.026), and CPI (F(1, 711) = 45.663, *p* < 0.001, η^2^ = 0.060), with small to medium effect sizes. In contrast with the previous t-tests, no statistically significant differences were found for VCI, PRI, FSIQ, or GAI, although trends in VCI and GAI approached significance.

#### 3.6.3. Effect of Age Group

Significant differences by age group (children vs. adolescents) were observed for VCI (F(1, 711) = 7.190, *p* = 0.008, η^2^ = 0.010), FSIQ (F(1, 711) = 4.375, *p* = 0.037, η^2^ = 0.006), and GAI (F(1, 711) = 5.673, *p* = 0.017, η^2^ = 0.008), with younger participants outperforming adolescents; also aligning with the previous t-tests. These effects were small in magnitude. No significant differences were found in PRI, WMI, PSI, or CPI.

## 4. Discussion

This study aimed to characterise the specific cognitive profile of children and adolescents diagnosed with ADHD through the implementation of the WISC-IV assessment. The results obtained demonstrate high ecological validity, due to the use of a tool that is widely applied in clinical and educational settings, as well as the inclusion of a large and representative sample (*n* = 719). Through the results, it may be concluded that the cognitive profiles observed in the clinical group (Figure 1) are mainly characterised by deficits in WMI, PSI, and CPI, thus confirming previous findings in the literature ([32]; [25]; [38]; [47]; [50]; [51]).

The deficits observed specifically in WMI and PSI are directly related to neurobiological alterations widely documented in the literature. In particular, the dorsolateral prefrontal cortex and the anterior cingulate cortex play essential roles in working memory and attentional regulation ([42]), while the basal ganglia and the limbic system are fundamental for inhibitory control and processing speed ([8]; [36]). These brain structures, consistently implicated in ADHD, may explain the specific cognitive difficulties identified in the present study.

On the other hand, the initial Student’s t-test analyses revealed significant group differences in VCI, PRI, and GAI; however, the MANOVA analysis enabled a more precise examination of these findings by controlling for the effects of variables such as age and sex. This approach confirmed that no significant differences exist in those indices, while reaffirming statistically significant differences in WMI, PSI, and CPI, with small to moderate effect sizes (η^2^ = 0.055, 0.026, and 0.060, respectively). These results reinforce the specificity of these domains as cognitive markers of ADHD. They are consistent with previous studies suggesting that ADHD does not involve a generalised impairment across all cognitive domains, but rather specific deficits in targeted cognitive functions ([24]; [38]; [50]).

Regarding sex differences, the descriptive data from this study reveal a significant disparity in the proportion of males and females diagnosed with ADHD. As shown in the results, the clinical group was predominantly composed of males (81.5%) compared to females (18.5%), with a statistically significant difference (χ^2^ = 12.030, *p* = 0.001). This finding is consistent with previous studies reporting a higher prevalence of ADHD in males ([6]). However, this disproportional representation may also indicate that females are often underdiagnosed ([57]; [42]), not only because they tend to exhibit differences in symptomatology, but also because they are less likely to be recognised as having ADHD by parents, teachers, or clinicians, especially in the absence of disruptive behaviour.

Moreover, Student’s t-test results showed that females with ADHD exhibited significantly higher PS scores than males with ADHD, with a moderate effect size (*p* < 0.001; d = 0.517). These findings were confirmed in the MANOVA through the interaction effect of sex and ADHD diagnosis, which remained significant for PSI even after controlling for age (F(1, 711) = 4.60, *p* = 0.032, η^2^ = 0.006). This supports previous research indicating substantial differences in the clinical manifestation of ADHD between males and females ([1]; [50]; [57]). Females tend to display more internalising symptoms, such as anxiety and emotional difficulties, and fewer externalising behaviours like hyperactivity and impulsivity, which may account for relatively better performance in tasks that demand speed and accuracy ([57]). However, this cognitive advantage may only contribute to a more accurate ADHD diagnosis in females when the assessment includes standardised psychometric tools such as the WISC-IV. Otherwise, their symptoms may go unnoticed in evaluations based solely on observable behavioural criteria, potentially leading to underdiagnosis due to both their subtler symptom profile and the influence of observer bias.

Regarding age differences, adolescents with ADHD (12–16 years) showed lower scores across all WISC-IV indices compared to children (6–11 years). T-test analyses revealed statistically significant differences with small effect sizes in the VCI (d = 0.437), PRI (d = 0.406), and their composite score, GAI (d = 0.479). These findings contrast with previous literature that suggested that certain cognitive deficits, particularly working memory, may improve with age ([6]; [23]; [43]; [55]). However, a similar developmental trend was also observed in the control group, where younger participants outperformed adolescents on the same indices (VCI: *p* = 0.003, d = 0.400; GAI: *p* = 0.011, d = 0.355; FSIQ: *p* = 0.005, d = 0.386). This pattern suggests that the decline in cognitive performance during adolescence may not be specific to ADHD, but could reflect general developmental factors affecting all adolescents, such as increasing academic, social, or emotional demands.

To further examine these effects, a factorial MANOVA (Table 6) was conducted to assess the main and interaction effects of age, diagnosis, and sex. The main effects of age on VCI and GAI remained statistically significant after controlling for sex and diagnosis, although effect sizes were small (η^2^ = 0.010 and η^2^ = 0.008, respectively). These results confirm that VCI and GAI are more sensitive to general developmental influences, such as age and schooling, rather than to ADHD-specific deficits. Crucially, however, the MANOVA did not reveal a significant interaction between age and ADHD diagnosis for any of the other cognitive indices, including WMI, PSI, or CPI. This pattern suggests that while certain cognitive domains like VCI and GAI may decline modestly with age across both groups, the specific deficits associated with ADHD, particularly WMI, PSI and CPI, remain stable from childhood to adolescence, reinforcing their role as core cognitive markers of the disorder rather than age-sensitive vulnerabilities. Furthermore, and by contrast, sex did not contribute to differences in VCI or GAI, but did show significant effects in PSI, WMI, and CPI, where females outperformed males. As mentioned earlier, this may be related to the tendency for females with ADHD to present fewer externalising symptoms and relatively stronger performance in tasks demanding speed and accuracy, which could partly explain their advantage in indices of cognitive efficiency.

Lastly, discrepancies between our findings and previous studies may stem from methodological differences, such as the clinical recruitment of the ADHD group in this study. This may reflect unique socioeconomic or motivational characteristics of families who seek diagnostic services, which could, in turn, influence cognitive outcomes. Longitudinal studies are needed to clarify developmental trajectories and to explore how contextual factors interact with neuropsychological functioning across different age groups.

On the other hand, the results from the logistic regression model further highlight the potential of the WISC-IV as a practical, complementary tool in the clinical diagnosis of ADHD. This predictive model may allow for more accurate identification of cases with cognitive profiles compatible with the disorder, particularly in the areas of WM and PS, thus facilitating objective clinical decision-making and reducing sole reliance on subjective observations from parents and teachers.

### 4.1. Educational and Clinical Implications

The findings of this study have clear clinical and educational implications due to their high ecological validity. First, the consistent identification of cognitive deficits in WMI and PSI reinforces the WISC-IV as a complementary tool in differential diagnosis, helping to reduce reliance on subjective parent and teacher reported measures. This is particularly relevant considering the comorbidity and frequent underdiagnosis of ADHD, and highlights the importance of including cognitive assessments alongside behavioural criteria in clinical practice.

Second, these findings provide concrete foundations for designing individualised interventions both in clinical practices and educational settings. For example, targeted working memory training programmes could significantly enhance academic performance in students with ADHD, particularly in school tasks that require simultaneous and efficient information manipulation ([46]). Similarly, interventions based on adapted physical exercise could enhance processing speed, which is especially relevant in educational settings where tasks often demand quick and accurate responses ([58]).

Third, the observed significant cognitive differences between sexes emphasises the need to tailor educational and clinical interventions accordingly. Females with ADHD may particularly benefit from support in emotional regulation, anxiety management, and organisational self-regulation, areas that are frequently more affected in this population ([57]), as well as a neuropsychological focus on attention deficits. While males may benefit more from psychological interventions that address both attention and processing speed deficits, which can aid in controlling hyperactivity/impulsivity and attention deficit symptomology. On the other hand, school accommodations could include modifications in the time allocated for tests and academic tasks, reduction in environmental distractions, and the use of differentiated pedagogical strategies based on the cognitive deficits consistently found in WM and PS in ADHD children and adolescents.

Overall, this study underscores the importance of using objective neuropsychological assessments to guide more effective diagnosis and interventions tailored to the specific needs of each child or adolescent with ADHD, which can significantly enhance educational, clinical, and psychosocial outcomes for this population.

### 4.2. Limitations and Future Research

Among the most relevant limitations of the present study is the lack of longitudinal data, which prevents the establishment of clear causal relationships regarding the temporal evolution of the identified cognitive deficits. Additionally, the study did not differentiate between the clinical subtypes of ADHD (inattentive, hyperactive/impulsive, or combined), which could have provided greater specificity to the findings.

Furthermore, the clinical group contained a higher proportion of males and was slightly younger on average than the control group. Although sex and age were statistically controlled in the analyses, these imbalances may reduce sensitivity to detect sex-specific effects and limit the generalisability of the findings, particularly to females. In addition, the sample was drawn from schools and clinical services in specific Spanish regions, which may restrict the applicability of results to other cultural or educational contexts. Future studies should aim to include a more balanced and diverse sample to strengthen external validity.

Another limitation concerns the measurement of general intellectual ability. While the WISC-IV provides composite scores such as the FSIQ and the GAI, these are not equivalent to the latent construct of general intelligence (*g*). FSIQ and GAI are derived from selected subtests, whereas *g* reflects the shared variance across a broader range of cognitive processes. Future research using structural equation modelling (SEM) could clarify whether the deficits observed here in WM and PS are independent of *g* or reflect more general intellectual impairments, thereby refining both theoretical models and diagnostic practice.

Future research should take these methodological aspects into account to broaden and refine the results obtained. Incorporating longitudinal designs, distinguishing ADHD subtypes, and applying SEM to examine the interplay between g and specific processes would allow to continue clarifying the neurocognitive profile of ADHD.

## 5. Conclusions

This study identified specific deficits in working memory, processing speed, and cognitive proficiency in children and adolescents with ADHD, which remained stable across age and were confirmed through multivariate analysis. Sex differences also emerged, with females showing better performance in processing speed. These findings support the use of the WISC-IV as a complementary tool for a more objective and thorough diagnosis.

## Figures and Tables

**Figure 1 behavsci-15-01279-f001:**
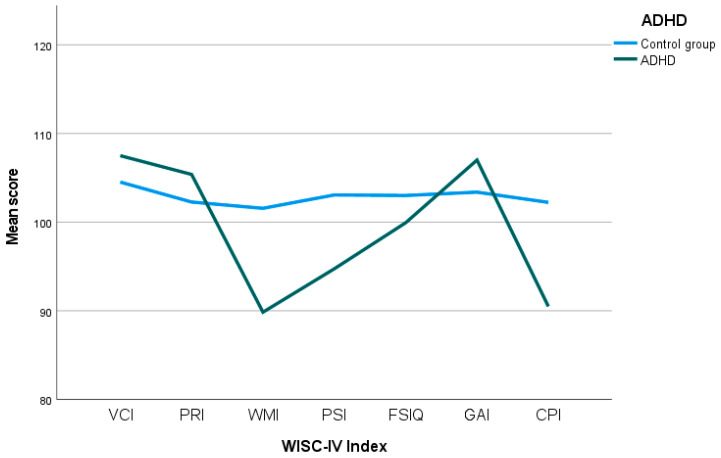
Comparison of mean WISC-IV index scores between the ADHD and control groups. VCI: Verbal Comprehension Index; PRI: Perceptual Reasoning Index; WMI: Working Memory Index; PSI: Processing Speed Index; FSIQ: Full-scale Intelligence Quotient; GAI: Global Ability Index; CPI: Cognitive Proficiency Index.

**Figure 2 behavsci-15-01279-f002:**
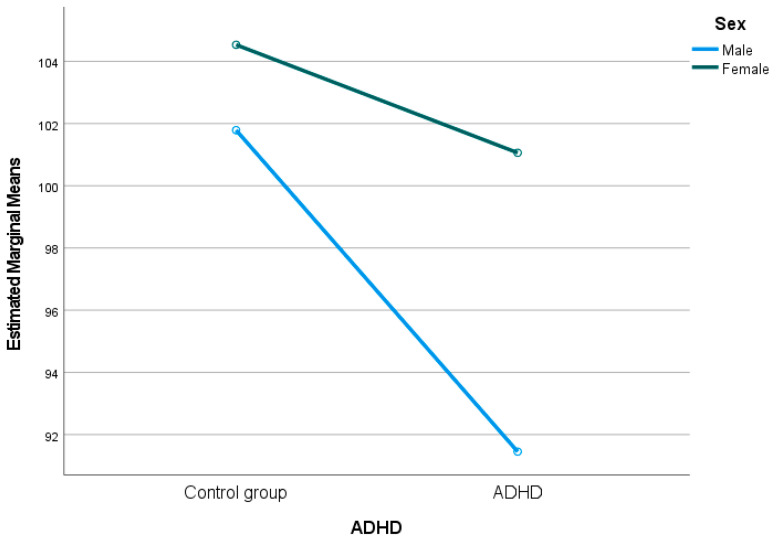
Estimated marginal means of PSI by sex and diagnostic group. Females with ADHD outperformed males with ADHD in PSI scores.

**Table 1 behavsci-15-01279-t001:** Descriptive Characteristics of the Sample.

	Control Group	Clinical Group	Total Sample
	*n* = 356	*n* = 363	*n* = 719
Age, M (SD)	8.87 (2.67)	8.17 (2.27)	8.51 (2.50)
Sex	*n* (%)	*n* (%)	*n* (%)
Males	251 (70.5%)	296 (81.5%)	547 (76.1%)
Females	105 (29.5%)	67 (18.5%)	172 (23.9%)
Age Group	*n* (%)	*n* (%)	*n* (%)
Children	284 (79.8%)	328 (90.4%)	612 (85.1%)
Adolescents	72 (20.2%)	35 (9.6%)	107 (14.9%)

Note. M: Mean; SD: Standard Deviation.

**Table 2 behavsci-15-01279-t002:** Comparison of WISC-IV Index Scores Between Control and Clinical Groups.

	Control Group M, (SD)	Clinical Group M, (SD)	*t*	*p*	d
Verbal Comprehension Index (VCI)	104.52 (14.57)	107.50 (13.07)	−2.887	0.004 *	0.21
Perceptual Reasoning Index (PRI)	102.27 (14.83)	105.39 (13.69)	−2.926	0.004 *	0.22
Working Memory Index (WMI)	101.56 (13.72)	89.85 (13.90)	11.372	<0.001 **	0.85
Processing Speed Index (PSI)	103.07 (12.39)	94.74 (13.90)	8.4	<0.001 **	0.63
Full-Scale IQ (FSIQ)	103.01 (14.25)	99.93 (13.08)	3.018	0.003 **	0.23
General Ability Index (GAI)	103.39 (14.98)	107.01 (13.34)	−3.424	0.001 **	0.25
Cognitive Proficiency Index (CPI)	102.23 (13.83)	90.50 (12.00)	12.161	<0.001 **	0.91

Note. M: Mean; SD: Standard Deviation; *t*: Student’s *t*-statistic; *p*: Significance level; d: Cohen’s d effect size. * *p* < 0.05; ** *p* < 0.01.

**Table 3 behavsci-15-01279-t003:** Comparison of WISC-IV Index Scores by Sex in the Control and ADHD Groups.

	Control Group					ADHD Group				
	Males M, (SD)	Females M, (SD)	*t*	*p*	d	Males M, (SD)	Females M, (SD)	*t*	*p*	d
Verbal Comprehension Index (VCI)	105.01 (14.94)	103.36 (13.65)	0.972	0.332	0.115	107.67 (13.46)	106.75 (11.24)	0.523	0.601	0.074
Perceptual Reasoning Index (PRI)	102.51 (14.82)	101.71 (15.67)	0.459	0.647	0.052	105.45 (13.82)	105.12 (13.15)	0.176	0.860	0.024
Working Memory Index (WMI)	101.05 (13.22)	102.80 (14.85)	−1.099	0.272	0.124	89.62 (14.07)	90.87 (13.16)	−0.663	0.508	0.092
Processing Speed Index (PSI)	103.01 (12.55)	103.23 (12.05)	−0.153	0.878	0.018	93.44 (13.78)	100.49 (13.08)	−3.819	<0.001 **	0.525
Full-Scale IQ (FSIQ)	103.20 (13.96)	102.55 (14.98)	0.393	0.695	0.045	99.64 (13.23)	101.22 (12.40)	0.894	0.372	0.121
General Ability Index (GAI)	103.59 (15.26)	102.90 (14.35)	0.393	0.695	0.047	107.10 (13.44)	106.61 (12.99)	0.269	0.788	0.037
Cognitive Proficiency Index (CPI)	101.89 (13.38)	103.05 (14.88)	−0.718	0.473	0.082	89.73 (11.93)	93.88 (11.83)	−2.574	0.010 **	0.349

Note. M: Mean; SD: Standard Deviation; *t*: Student’s *t*-statistic; *p*: Significance level; d: Cohen’s d effect size. ** *p* < 0.01.

**Table 4 behavsci-15-01279-t004:** Comparison of WISC-IV Index Scores by Age in the Control and ADHD Groups.

	Control Group					ADHD Group				
	Children (6–11 Years)	Adolescents (12–16 Years)	*t*	*p*	d	Children (6–11 years)	Adolescents (12–16 Years)	*t*	*p*	d
	M, (SD)	M, (SD)				M, (SD)	M, (SD)			
Verbal Comprehension Index (VCI)	105.68 (14.44)	99.96 (14.30)	3.099	0.003 **	0.400	108.04 (13.01)	102.43 (12.66)	2.433	0.015 *	0.437
Perceptual Reasoning Index (PRI)	103.00 (15.42)	99.42 (11.91)	1.836	0.067	0.260	105.92 (13.61)	100.40 (13.60)	2.280	0.023 *	0.406
Working Memory Index (WMI)	102.14 (14.41)	99.29 (10.35)	1.577	0.116	0.227	89.86 (13.81)	89.77 (14.85)	0.034	0.973	0.006
Processing Speed Index (PSI)	103.47 (12.69)	101.50 (11.04)	1.207	0.228	0.166	94.91 (13.83)	93.11 (14.65)	0.728	0.467	0.126
Full-Scale IQ (FSIQ)	104.07 (14.50)	98.85 (12.46)	2.803	0.005 **	0.386	100.44 (13.02)	95.17 (12.80)	2.280	0.023 *	0.405
General Ability Index (GAI)	104.40 (15.35)	99.39 (12.75)	2.555	0.011 *	0.355	107.60 (13.35)	101.49 (12.12)	2.597	0.010 *	0.479
Cognitive Proficiency Index (CPI)	102.88 (14.55)	99.69 (10.19)	1.749	0.081	0.254	90.63 (11.92)	89.31 (12.77)	0.614	0.540	0.107

Note. M: Mean; SD: Standard Deviation; *t*: Student’s *t*-statistic; *p*: Significance level; d: Cohen’s d effect size. * *p* < 0.05; ** *p* < 0.01.

**Table 5 behavsci-15-01279-t005:** Binary Logistic Regression Predicting Group Classification Based on WISC-IV Indices.

	B	SE	Wald	*p*	OR	95% C.I. LL–UL
VCI	0.054	0.008	43.636	<0.001	1.056	1.04–1.07
PRI	0.058	0.008	49.534	<0.001	1.060	1.04–1.08
WMI	−0.105	0.010	120.150	<0.001	0.901	0.88–0.92
PSI	−0.057	0.008	53.011	<0.001	0.945	0.93–0.96
Constant	3.881	0.980	15.673	<0.001	48.452	

Note. B = Regression coefficient; SE = Standard error; *p* = significance level; OR = Odds Ratio; C.I. = 95% confidence interval; LL = lower limit; UL = upper limit. VCI: Verbal Comprehension Index; PRI: Perceptual Reasoning Index; WMI: Working Memory Index; PSI: Processing Speed Index.

**Table 6 behavsci-15-01279-t006:** Main Effects of Sex, Diagnostic Group, and Age Group on WISC-IV Indices.

	Index	F(1,711)	*p*	η^2^
Sex	VCI	0.230	0.631	0.000
	PRI	1.528	0.271	0.002
	WMI	4.382	0.037 *	0.006
	PSI	14.902	0.000 **	0.021
	FSIQ	4.971	0.026 *	0.007
	GAI	1.363	0.243	0.002
	CPI	12.962	0.000 **	0.018
Group (ADHD vs. Control)	VCI	3.466	0.063	0.005
	PRI	1.886	0.170	0.003
	WMI	41.476	0.000 **	0.055
	PSI	18.619	0.000 **	0.026
	FSIQ	2.785	0.096	0.004
	GAI	3.508	0.061	0.005
	CPI	45.663	0.000 **	0.060
Age Group (Children vs. Adolescents)	VCI	7.190	0.008 *	0.010
	PRI	2.959	0.086	0.004
	WMI	0.014	0.906	0.000
	PSI	0.246	0.620	0.000
	FSIQ	4.375	0.037 *	0.006
	GAI	5.673	0.017 *	0.008
	CPI	0.139	0.710	0.000

Note. η^2^ = Partial eta squared. * *p* < 0.05, ** *p* < 0.001. VCI = Verbal Comprehension Index; PRI = Perceptual Reasoning Index; WMI = Working Memory Index; PSI = Processing Speed Index; FSIQ = Full-Scale Intelligence Quotient; GAI = General Ability Index; CPI = Cognitive Proficiency Index.

## Data Availability

The data presented in this study are available on request from the corresponding author. The data are not publicly available due to privacy and ethical restrictions, as participants are minors.

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
