# Peer review of "Cognitive Profiling of Children and Adolescents with ADHD Using the WISC-IV"

_behavsci, 2025, doi:10.3390/bs15091279_

Round 1
Reviewer 1 Report
Comments and Suggestions for Authors
Reviewer Comments on Manuscript ID: behavsci-3787360
In this study, the authors examined the relationship between ADHD diagnosis and intelligence in 719 children and adolescents in Spain. Participants completed the Wechsler Intelligence Scale for Children – 4th Edition. MANOVA analysis showed that, after controlling for sex and age group, the ADHD group had higher scores in working memory, processing speed, and cognitive proficiency, but not in full-scale IQ, verbal comprehension index, perceptual reasoning, or general ability. I have several concerns that should be addressed to improve the clarity, rigor, and interpretability of the manuscript.
Introduction:
- In the literature review, are there prior similar studies comparing IQ between participants with ADHD and those without ADHD? How does your study extend the findings of previous research?
- On line 182 (page 4), please correct the sentence: “he main objective of the present study is.”.
- On page 4, after stating your research aim, it would be helpful to add clear predictions or hypotheses.
Method:
- On page 5, how many participants met the inclusion criteria for your study?
- On pages 5–6, were ADHD symptoms reported by both parents and teachers? How did you handle inconsistencies between their responses? Why did you not use diagnostic assessments conducted by clinical professionals in healthcare centers?
- For Section 2.5 (Statistical Analysis), it would be helpful to include a brief summary of the analyses conducted in the study and the software used at the beginning of this section.
Results:
- On page 7, why is there a noticeably higher number of male participants compared to females in the sample? Please clarify this imbalance, as the unbalanced gender ratio may limit the generalizability of your findings. It would also be appropriate to include this as a limitation in the discussion.
- On page 7, could you clarify whether there were any significant age differences between the clinical and control groups?
Discussion:
- In the discussion, please explain why sex, group, and age group differences were observed in some IQ index but not in others, based on Table 8.
- Add a discussion of the implications of your findings for future research and practical applications.
Author Response
Reply to Reviewer 1:
Dear reviewer,
We greatly appreciate the feedback you provided regarding our study. We will address each concern and hope that our clarifications and the revisions made to the manuscript adequately respond to your comments. We believe these changes have helped us improve the clarity, rigour, and overall contribution of the paper. Please find below a detailed, point-by-point response.
Introduction:
- In the literature review, are there prior similar studies comparing IQ between participants with ADHD and those without ADHD? How does your study extend the findings of previous research?
We thank the reviewer for this observation. Previous studies using the WISC-IV, such as Fenollar-Cortés et al. (2015), Navarro-Soria et al. (2020), and Thaler et al. (2013), compared IQ and core indices between children with ADHD and control groups, consistently reporting significant deficits in working memory and processing speed, while verbal comprehension and perceptual reasoning tended to be preserved. We have clarified this point in lines 180-184. The present study extends this body of research by relying on a much larger sample (n = 719) and by examining differences not only between diagnostic groups but also across sex and age groups, applying additional statistical procedures such as logistic regression and factorial MANOVA. These methodological improvements provide a more comprehensive characterisation of cognitive patterns in ADHD and allow for the identification of interaction effects that were not addressed in earlier studies. As a result, our findings both confirm and refine previous evidence, offering a broader perspective that is further discussed in the Results and Discussion sections.
- On line 182 (page 4), please correct the sentence: “he main objective of the present study is.”.
Thank you for noting this correction. The typo has been corrected, along with others identified during proofreading.
- On page 4, after stating your research aim, it would be helpful to add clear predictions or hypotheses.
We agree with the reviewer and have now added explicit hypotheses at the end of the introduction.
Method:
- On page 5, how many participants met the inclusion criteria for your study?
We appreciate the reviewer’s observation. Initially, we did not include this information because of the complexity of handling such a large dataset and the overlap of cases from different recruitment sources, which made it difficult to report exclusions in detail within the main text. However, we fully agree that this information strengthens the transparency of the sample selection. Therefore, we have now specified in the Participants section (lines 237–241) the number of participants that did not meet the inclusion criteria.
- On pages 5–6, were ADHD symptoms reported by both parents and teachers? How did you handle inconsistencies between their responses? Why did you not use diagnostic assessments conducted by clinical professionals in healthcare centers?
Yes, ADHD symptoms were reported by both parents and teachers using standardised questionnaires. In cases of inconsistency, a semi-structured interview with parents was conducted for clarification, and when needed, an additional review with the participant’s teacher was carried out (lines 241-245). Moreover, as described in lines 269-273, the research team checked all scores for internal consistency. Participants whose assessments remained ambiguous or inconsistent after this process were excluded to maintain methodological rigour.
Regarding diagnosis, all children in the ADHD group had an initial diagnosis made by qualified clinical professionals in healthcare centres (lines 263-264). To ensure that all participants were assessed with the same procedures, standardised instruments (SDQ, ADHD-RS-IV, and Conners-3) were subsequently administered by our trained team, which allowed us to verify and complement the clinical diagnosis while ensuring comparability across participants.
- For Section 2.5 (Statistical Analysis), it would be helpful to include a brief summary of the analyses conducted in the study and the software used at the beginning of this section.
We agree that it is clearer for the reader to present this information at the beginning of the section rather than at the end. You can see the changes in lines 283-286.
Results:
- On page 7, why is there a noticeably higher number of male participants compared to females in the sample? Please clarify this imbalance, as the unbalanced gender ratio may limit the generalizability of your findings. It would also be appropriate to include this as a limitation in the discussion.
We thank the reviewer for this observation. As shown in the descriptive results (Section 3.1, page 7), there was a significantly higher proportion of males in the ADHD group. This imbalance is consistent with epidemiological findings indicating that ADHD is approximately twice as prevalent in boys compared to girls. Nevertheless, we acknowledge that the overrepresentation of males in our sample may limit the generalisability of our findings to females with ADHD, and we have now noted this as a limitation in Limitations and Future Research section, lines 583-584.
- On page 7, could you clarify whether there were any significant age differences between the clinical and control groups?
We thank the reviewer for pointing this out. You are correct that the difference in age distribution between groups is important to report, and we regret not having included this information in the initial submission. We have now added the chi-square result in the Results section (lines 355-357): χ²(1) = 15.891, p = .000, indicating that the ADHD group contained a lower proportion of adolescents compared to the control group.
Importantly, this imbalance does not affect the validity of our findings, as age was already taken into account in the analyses. Specifically, the sample was dichotomized into children (6–11 years) and adolescents (12–16 years), and age group was included as a factor in the MANOVA. Both approaches confirmed that the ADHD-specific deficits in working memory, processing speed, and cognitive proficiency were stable across age groups. To ensure transparency, we have also acknowledged this imbalance in the Limitations and Future Research section, lines 611-617.
Discussion:
- In the discussion, please explain why sex, group, and age group differences were observed in some IQ index but not in others, based on Table 8.
Yes, based on table 8 (now table 6), there were some indices more or less sensitive to sex, group or age group, you can see we have clarified this in lines 520-536. For VCI and GAI, the differences were small and mainly linked to general age-related changes across all participants, not specifically to ADHD. By contrast, the core ADHD-related difficulties were consistently observed in WMI, PSI, and CPI, and these remained significant regardless of age or sex. Importantly, sex differences did not affect VCI or GAI, but did play a role in PSI (and to a lesser extent WMI and CPI), where females outperformed males. Finally, the MANOVA confirmed there were no age x diagnosis interactions, meaning the ADHD-specific deficits in WMI, PSI, and CPI were stable from childhood to adolescence.
- Add a discussion of the implications of your findings for future research and practical applications.
Thank you for this suggestion, we have expanded the limitation section to “Limitations and Future Research”. You can see the additions in lines 611-630. The practical implications, being both clinical and educational, have been expanded and made into a section of its own, as section 4.1 Educational and Clinical Implications.

Reviewer 2 Report
Comments and Suggestions for Authors
Thank you for the opportunity to review the article titled - Cognitive Profiling of Children and Adolescents with ADHD using the WISC-IV. I have some revision suggestions for your consideration. To start with some minor issues, in the title 'using' shoud be capitalized? Similarly, there are some typos in the manuscript, e.g., p.4 -line 182 'he main objective' should be 'the main objective'?
Secondly, the core objective is to find an more objective approach to identify or inform the early identification of ADHD. Given there are so many statistical analyses used, it is important to list the research questions in the introduction. Also, in the methods section, please elaborate how each statistical analysis was adopted to answer each respective research question.
Third, in the Results section, some sections perhaps could be more condensed. E.g., 3.3.1 & 3.3.2; as well as Tables 3 & 4 (can be combined into one Table). Similarly, 3.4.1 & 3.4.2; as well as Tables 5 & 6.
Fourth, given there are more males in the clinical group, I think sex should be controlled for in the logistic regression.
Fifth, it is unclear to me about the purpose of MANOVA analysis. What additional information does this analysis aim to offer?
Sixth, the limitations section is quite limited; please expand it into "Limitations & Future Research Directions" with more elaborations.
Last but not least, it is important for the authors to double check if any previous studies along this line were conducted. If not, please make it explicit in your introduction.
Thank you for your attention to these issues.
Author Response
Reply to Reviewer 2:
Dear reviewer,
We greatly appreciate the feedback you provided regarding our study. We will address each concern and hope that our clarifications and the revisions made to the manuscript adequately respond to your comments. We believe these changes have helped us improve the clarity, rigour, and overall contribution of the paper. Please find below a detailed, point-by-point response.
- The theoretical framework in which the study is embedded is limited. There is both classical and recent theoret ical research in developmental psychopathology that is not even mentioned. However, the field would profit if this work is used to generate predictions that would be tested by this study. For instance, in what processes would alternative theories predict differences in the processes studied here? When these differences would appear in age? Some references are given below:
We thank the reviewer for this valuable suggestion. We agree that embedding our work more explicitly within the framework of developmental psychopathology strengthens the theoretical basis of the study. In response, we have integrated references to both classical and recent contributions in this field. These additions can be found in the introduction (lines 45-50, 134-136, 147-150, and 194-197).
- The paper would profit if more systematic analyses would be conducted. For instance, I would suggest that structural equation modelling is used to examine if there are differences between the two groups in g as contrasted to specific processes as those identified here. The theories above predict that there must be differences in both g and specific processes, but this cannot be identified with the methods used here.
Thank you very much for this suggestion. We fully agree that structural equation modelling would provide a valuable framework for examining the interplay between general intelligence and specific cognitive processes in ADHD. However, conducting SEM analyses was beyond the scope of the present study, which focused on comparing WISC-IV index scores and their predictive utility using classical statistical methods. That said, we recognise the importance of your suggestion. We have therefore explicitly acknowledged this limitation and highlighted SEM as a promising direction for future research in the revised “Limitations and Future Research” section, lines 619-625.
- The educational and clinical implications of the study must be systematically discussed. In the present version of the paper this is avoided.
Thank you for this valuable comment. In the revised manuscript, we have reorganised the paragraph on implications into a new subsection (4.2. Educational and Clinical Implications). The content has been structured systematically, with each main finding (WMI and PSI deficits, stability across age, and sex differences) explicitly linked to corresponding clinical and educational applications.

Reviewer 3 Report
Comments and Suggestions for Authors
This study searched for differences between children with ADHD and typically developing children and adolescents in processes assessed by the Wechsler Intelligence Scale for Children (WISC-IV). It is reported that that the ADHD children scored lower in Working Memory, Processing Speed, and Cognitive Proficiency but similarly in Verbal Comprehension and Perceptual Reasoning. It was also found that Girls with ADHD outperformed boys in Processing Speed. This is an interesting study that might be potentially useful to be available. However, it has several limitations that must be removed before it is accepted.
- The theoretical framework in which the study is embedded is limited. There is both classical and recent theoretical research in developmental psychopathology that is not even mentioned. However, the field would profit if this work is used to generate predictions that would be tested by this study. For instance, in what processes would alternative theories predict differences in the processes studied here? When these differences would appear in age? Some references are given below:
Cicchetti, D., & Cohen, D. J. (1995). Perspectives on developmental psychopathology. In Cicchetti, D., & Cohen, D. J. (Eds.), Developmental psychopathology: Theory and method. vol. 1, (pp. 3–20). Wiley.
Cicchetti, D., & Tucker, D. (1994). Development and self-regulatory structures of the mind. Development and Psychopathology, 6(4), 533–549.
Demetriou, A., Spanoudis, G., & Papadopoulos, T. C. (2025). The typical and atypical developing mind: A common model. Development and Psychopathology, 37(2), 1095-1107.
- The paper would profit if more systematic analyses would be conducted. For instance, I would suggest that structural equation modeling is used to examine if there are differences between the two groups in g as contrasted to specific processes as those identified here. The theories above predict that there must be differences in both g and specific processes, but this cannot be identified with the methods used here.
- The educational and clinical implications of the study must be systematically discussed. In the present version of the paper this is avoided.
Author Response
Reply to Reviewer 3:
Dear reviewer,
We greatly appreciate the feedback you provided regarding our study. We will address each concern and hope that our clarifications and the revisions made to the manuscript adequately respond to your comments. We believe these changes have helped us improve the clarity, rigour, and overall contribution of the paper. Please find below a detailed, point-by-point response.
1. To start with some minor issues, in the title 'using' should be capitalized? Similarly, there are some typos in the manuscript, e.g., p.4 -line 182 'he main objective' should be 'the main objective'?
We thank the reviewer for noting these corrections. The typos have been corrected, along with others identified during proofreading.
2. Secondly, the core objective is to find a more objective approach to identify or inform the early identification of ADHD. Given there are so many statistical analyses used, it is important to list the research questions in the introduction. Also, in the methods section, please elaborate how each statistical analysis was adopted to answer each respective research question.
We agree that including the research questions in the introduction improves clarity. These have been incorporated in lines 190-195 to enhance the fluency of the text and to provide a clearer transition to the Methods section. Each statistical analysis has also been linked to each RQ, you can see the changes in the Statistical Analysis section.
3. Third, in the Results section, some sections perhaps could be more condensed. E.g., 3.3.1 & 3.3.2; as well as Tables 3 & 4 (can be combined into one Table). Similarly, 3.4.1 & 3.4.2; as well as Tables 5 & 6.
Thank you for the recommendation. We have combined Tables 3 and 4 into a single table of sex-based comparisons (now Table 3), and Tables 5 and 6 into a single table of age-based comparisons (now Table 4), while maintaining separate subsections 3.3 and 3.4 for clarity.
4. Fourth, given there are more males in the clinical group, I think sex should be controlled for in the logistic regression.
Thank you very much for this comment. We fully agree that the unequal distribution of sex should be addressed in the logistic regression analysis. Following your suggestion, we repeated the analysis controlling for sex, and also included age, given the slight differences between groups. The results were highly consistent with those of the original model, the same WISC-IV indices emerged as significant predictors, and the overall accuracy of the model was practically unchanged (original model: Nagelkerke R² = .44, 78% correctly classified; controlled model: Nagelkerke R² = .46, 79% correctly classified). This indicates that the predictive capacity of the WISC-IV indices is not driven by demographic imbalances. However, we believe it is still important to highlight this in the manuscript, so we have added this clarification in lines 437-440.
Furthermore, upon revising this section, we also noticed that some relevant details had not been included in the initial submission (the model chi-square test, percentage of correctly classified cases, and the regression constant). These have now been added to the manuscript to provide a clearer description of the logistic regression results.
5. Fifth, it is unclear to me about the purpose of MANOVA analysis. What additional information does this analysis aim to offer?
Thank you for your question, it is important to note that while t-tests and logistic regression provided valuable information about group differences and predictive utility, the MANOVA was applied because it allows the simultaneous examination of diagnostic group, sex, and age group across all WISC-IV indices, and therefore can identify both main effects and interactions that cannot be captured by univariate tests alone.
While some of the MANOVA findings, such as the sex x diagnosis interaction on PSI, aligned with the direction of the t-test results, the MANOVA added value by confirming that this effect remained significant when diagnosis, sex, and age were considered simultaneously, and by clarifying that the interaction was specific to the clinical group. In addition, the MANOVA confirmed that the most robust differences between ADHD and control groups were concentrated in WMI, PSI, and CPI, whereas differences in VCI, PRI, and GAI observed in t-tests did not remain significant once sex and age were included in the model. Finally, the MANOVA showed that age effects were limited to VCI, FSIQ, and GAI, while no effects of age were found on WMI or PSI, confirming that the small age difference between groups does not explain the core ADHD deficits.
6. Sixth, the limitations section is quite limited; please expand it into "Limitations & Future Research Directions" with more elaborations.
Thank you, we have expanded this section, see additions in lines 611-630.
7. Last but not least, it is important for the authors to double check if any previous studies along this line were conducted. If not, please make it explicit in your introduction.
We thank the reviewer for this observation. Previous studies using the WISC-IV, such as Fenollar-Cortés et al. (2015), Navarro-Soria et al. (2020), and Thaler et al. (2013), compared IQ and core indices between children with ADHD and control groups, consistently reporting significant deficits in working memory and processing speed, while verbal comprehension and perceptual reasoning tended to be preserved. We have clarified this point in lines 180-184.

Round 2
Reviewer 2 Report
Comments and Suggestions for Authors
Thank you for addressing the comments carefully. One minor suggestion is to double check the consistency use of females vs. girls (e.g., in the hypotheses part).
Author Response
Thank you very much for this observation. We have carefully revised the manuscript to ensure consistent terminology, and now the term females is used uniformly throughout the text. .

Reviewer 3 Report
Comments and Suggestions for Authors
The paper has improved considerably. It is useful to be available both for theoretical and practical reasons. Therefore, I propose that it is accepted for publication.
Author Response
We sincerely appreciate the time and effort you invested in reviewing our manuscript. We are grateful for your constructive feedback throughout the process and for recommending our article for publication.
